# PRAME Immuno-Expression in Cutaneous Sebaceous Carcinoma: A Single Institutional Experience

**DOI:** 10.3390/jcm11236936

**Published:** 2022-11-24

**Authors:** Gerardo Cazzato, Anna Colagrande, Giuseppe Ingravallo, Teresa Lettini, Angela Filoni, Francesca Ambrogio, Domenico Bonamonte, Miriam Dellino, Carmelo Lupo, Nadia Casatta, Leonardo Resta, Eugenio Maiorano, Eliano Cascardi, Andrea Marzullo

**Affiliations:** 1Section of Pathology, Department of Emergency and Organ Transplantation (DETO), University of Bari “Aldo Moro”, 70124 Bari, Italy; 2Section of Dermatology, “Vito Fazzi” Hospital, 73100 Lecce, Italy; 3Section of Dermatology and Venereology, Department of Biomedical Sciences and Human Oncology (DIMO), University of Bari “Aldo Moro”, 70124 Bari, Italy; 4Section of Gynecology and Obstetrics, Department of Biomedical Sciences and Human Oncology (DIMO), University of Bari “Aldo Moro”, 70124 Bari, Italy; 5Innovation Department, Diapath S.p.A., Via Savoldini n. 71, 24057 Martinengo, Italy; 6Department of Medical Sciences, University of Turin, 10124 Turin, Italy; 7Pathology Unit, FPO-IRCCS Candiolo Cancer Institute, 10060 Candiolo, Italy

**Keywords:** PRAME, preferentially expressed antigen in melanoma, immunohistochemistry, differential diagnosis, sebaceous carcinoma, malignancy

## Abstract

Background: In recent years, great research interest has been directed to the diagnostic, therapeutic and marker role of Preferentially expressed Antigen in Melanoma (PRAME) in the setting of various human neoplasms. Although it has been extensively studied mainly in the differential diagnosis setting of melanocytic pigmented lesions, still very few papers have analyzed the usefulness or otherwise of PRAME in the context of other non-melanoma skin cancers (NMSC). (2) Methods: In this paper, we report the data of our experience of 21 cases of sebaceous carcinoma (SC) classified in the three WHO grade and collected in the period between January 2005 and 31 October 2022, on which immunostaining for PRAME was performed; Non-parametric Mann–Whitney test for non-normally distributed values was performed. A comparison was made of the means between the three study groups (grade I, II and III). A value of *p* ≤ 0.05 was set as statistically significant (3) Results: Only seven cases (33.3%) were positive with an immunoscore of 2+/3+ for intensity and 1+/2+ for percentage cells positivity, while 14 cases (66.6%) were totally or nearly totally negative for PRAME with a few of sebaceous-like cells positive with an immunoscore of 1+. Eight cases of SC grade I were immunostaining for PRAME, a level of the cytoplasm of foci of sebaceous differentiation with a significant statical value (*p* < 0.0001) with respect to ten cases of SC grade II; furthermore, the eight cases of grade I were positive for PRAME in the same areas respect the 3 cases of SC grade III (*p* = 0.0303). There were no statistical significance between the 10 cases of grade II and 3 cases of grade III (*p* = 0.2028); (4) Conclusions: PRAME not seems to add particular information in the case of histopathological diagnostics of SC where other markers, including adipophylline, can be quite indicative. It seems, on the other hand, that PRAME can be useful in the subclassification setting of sebaceous carcinoma in grades I–II–III according to the directives of the latest WHO 2018, highlighting the foci of mature sebaceous differentiation most present in grades 1–2 and almost completely absent in grade 3 of the SC.

## 1. Introduction

In recent years, great research interest has been directed to the diagnostic, therapeutic and marker role of Preferentially expressed Antigen in Melanoma (PRAME) in the setting of various human neoplasms [1,2]. First discovered and described in 1997 by Ikeda H. et al., in an autologous T lymphocytes clone in a patient with metastatic cutaneous melanoma, PRAME has returned to the attention of researchers both in the context of possible immunotherapies [3,4] and its diagnostic role in lesions mainly of a melanocytic nature [5,6]. More specifically, the literature studies report an important ability of PRAME to be positive in cases of Malignant Melanoma (MM) rather than in benign or dysplastic melanocytic lesions, although it seems important to consider PRAME as another indicator of the likely nature of a lesion rather than as clear and unambiguous evidence of benignity/malignancy of a pigmented lesion [7].

On the other hand, other studies have reported that immunostaining for PRAME can also be used in other neoplastic settings, including soft tissue sarcomas [8], non-small cell lung cancer [9], breast carcinoma [10], renal cell carcinoma [11], ovarian carcinoma [12], leukemia [13] and synovial sarcoma [14]. In this background, the importance and use of PRAME in non-melanoma skin cancer skin lesions have not yet been reported in the literature, with only a few works that have tried to shed light on the topic [15,16,17,18].

Sebaceous carcinoma (SC) is a sebaceous differentiation malignant tumor and is classified in the latest edition of the World Health Organization 2018 among the “skin adnexal tumors with sebaceous differentiation” [19]. Briefly, SC occurs mainly in middle-aged or elderly individuals and can affect any part of the body, with a predisposition for the head/neck district, particularly periocular location.

The histological grading suggested in this classification [19] proposes tumors with well-demarcated, roughly equally sized cellular lobules as grade I; SC with an admixture of well-defined nests with some infiltrative features and/or confluence of nests as grade II and, finally, SC with highly invasive growth and/or medullary sheet-like pattern are classified as grade III [19].

In this paper, we report the data of our experience of 21 cases of sebaceous skin cancer classified in the three WHO grade and collected in the period between January 2005 and 31 October 2022, on which immunostaining for PRAME was performed.

## 2. Materials and Methods

### 2.1. Case Selection

The selection of the cases was carried out through research on the computer system of the Pathological Anatomy laboratory of the University of Bari “Aldo Moro”, looking for the following words: “sebaceous carcinoma” or “adnexal sebaceous carcinoma” in a period of time included from 1 January 2015 to 31 December 2021. All cases of both ocular-type and extraocular sebaceous carcinoma were examined, and the slides were re-evaluated by two dermatopathologists (G.C. and A.C.); finally, only the cases with total agreement by both pathologists were included. Clinical and demographic information was retrieved from the cases studied.

### 2.2. PRAME Immunostaining and Assessment of Positivity

For the study of PRAME immuno-expression, 5-micron thick tissue sections were cut from formalin-fixed paraffin-embedded (FFPE) blocks, and the slides were all stained with anti-PRAME antibody Ab219650, rabbit monoclonal, in 1:250 dilution.

We categorized PRAME tumor cells’ percentage positivity and intensity of immunostaining in a cumulative score obtained by adding the quartile of positive tumor cells following as 0, 1+, 2+, 3+, 4+ to the PRAME expression intensity in tumor cells categorized as 0, 1+, 2+, 3+. More specifically, we used the following scores for the percentage positivity of tumor cells: 0% (score 0), 1% to 25% (score 1+), 26% to 50% (score 2+), 51% to 75% (score 3+) and 76% to 100% (score 4+). Furthermore, we used a score for intensity by measuring cytoplasm (not nuclear!) immunostaining for PRAME as weak, moderate, or strong (1+, 2+, or 3+, respectively). A composite score for each case was determined by adding the scores for the percentage and intensity of immunostaining. A composite score of 4 to 7 was considered positive, and a composite score of 0 to 3 was considered negative.

Sebaceous glands were used as an internal control to confirm the functioning of the PRAME antibody stain. These patterns of immuno-expression were estimated by dermatopathologists during the review of the cases.

### 2.3. Statistical Analysis

The mean and standard deviation values for the 10 fields were recorded for each case. Non-parametric Mann–Whitney test for non-normally distributed values was performed. A comparison was made of the means between the three study groups (grade I, II and III). A value of *p* ≤ 0.05 was set as statistically significant. All statistical analyses were made using the Prism 9.0.3 program, GraphPad Software, 9.4.2 version, 2021 (La Jolla, CA, USA).

## 3. Results

A total of 21 sebaceous carcinomas were included. Eleven patients were males (52.4%), and ten patients were females (47.6%). The median age was 74 years.

Six lesions were ocular-type (28.6%), eight lesions were on the head and neck (38.0%) and seven lesions were distributed in different districts of the body. Ten cases (47.6%) were subjected to surgery and re-excision, mostly in the cases of ocular-type neoplasms.

Clinical, topographical and treatment characteristics are summarized in Table 1.

In terms of WHO grading, 8 cases (38.0%) were classified as WHO grade I; 10 cases (47.6%) were classified as grade II and 4 cases (19.0%) were classified as grade III.

Regarding PRAME immunostaining, only 7 cases (33.3%) were positive with an immunoscore of 2+/3+ for intensity and 1+/2+ for percentage cells positivity (Figure 1A–C), while 14 cases (66.6%) were totally or nearly totally negative for PRAME with a few of sebaceous-like cells positive with an immunoscore of 1+ (Figure 1D).

In terms of grading WHO 2018, 8 cases of SC grade I were immunostaining for PRAME, a level of the cytoplasm of foci of sebaceous differentiation with a significant statical value (*p* < 0.0001) with respect to 10 cases of SC grade II; furthermore, the 8 cases of grade I were positive for PRAME in the same areas respect the 3 cases of SC grade III (*p* = 0.0303). There was no statistical significance between the 10 cases of grade II and 3 cases of grade III (*p* = 0.2028).

## 4. Discussion

Sebaceous carcinoma is a fairly characterized entity in the literature [20,21,22,23], but very few works have tried to elucidate the possible usefulness of PRAME in this setting. For example, the paper by Donnell et al. [23] examined the immunoexpression of PRAME and another marker, adipophilin, in 20 sebocyte neoplasia compared to 32 controls. The authors described strong PRAME immunostaining (both cytoplasmic and perinuclear) in 20/20 sebaceous neoplasia, with few differences in intensity and percentage of positivity. Furthermore, the authors reported a high PRAME positivity rate (15/16 cases) of sebaceous lesions with an extensive basaloid component. This finding is only partially confirmed by our study, in which we found almost total negativity at the level of the basaloid sebocytes but moderate to strong cytoplasmic positivity in the areas of mature sebaceous differentiation. In another paper almost cited by Elsensohn et al. [18], the authors conducted an elegant study on various types of non-melanoma skin cancer, including four cases of SC in which a cytoplasmic positivity for PRAME was described in the well-differentiated sebocytes, but there was only low-intensity nuclear expression in one of four cases, involving fewer than 25% of tumor cells. In our study, however, we never found nuclear positivity for PRAME. Furthermore, in a very recent paper of October 2022, Joanna K.M. et al. [15] described their experience with various lesions belonging to NMSCs, including 193 sebaceous lesions. The authors analyzed these lesions, obtaining, as a result, near negativity of the basaloid and mature sebocytes as compared to germinative sebocytes; moreover, although PRAME was positive in the foci of sebaceous differentiation, the authors reported a low specificity for sebaciomas and sebaceous carcinomas, similarly to what we also found in this paper.

## 5. Conclusions

In conclusion, although PRAME represents a promising marker not only for immunotherapeutic purposes but also for diagnostic/prognostic purposes, its usefulness is proven in the setting of melanocytic lesions, but it would not seem, from the existing studies in the literature, to which ours is added, that it is able to add particular information in the case of histopathological diagnostics of sebaceous carcinoma, where other markers, including adipophylline, can be quite indicative. It seems, on the other hand, that PRAME can be useful in the subclassification setting of sebaceous carcinoma in grades I–II–III according to the directives of the latest WHO 2018, highlighting the foci of mature sebaceous differentiation most present in grades 1–2 and almost completely absent in grade 3 of the SC. The major limitation of this study is the small number of cases of SC. Further studies with larger cases are needed to confirm or refute these results.

## Figures and Tables

**Figure 1 jcm-11-06936-f001:**
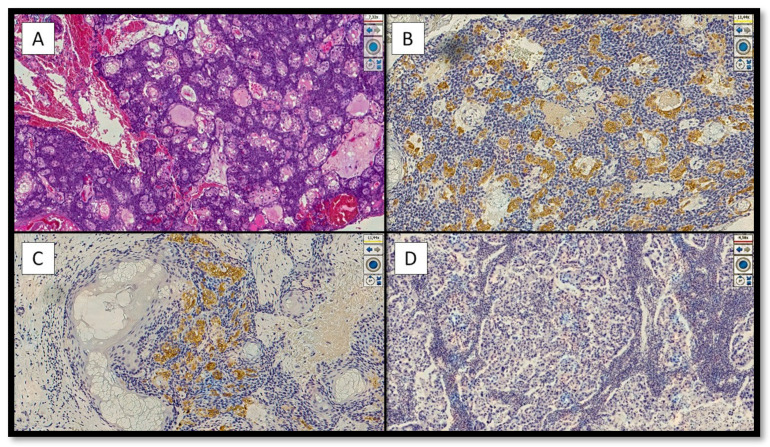
Examples of positivity and/or negativity for immunostaining of PRAME in the cases analyzed in this study. (**A**) Histological micrograph showing a sebaceous carcinoma of grade I in order to World Health Organization Skin Tumor Classification 2018, characterized by numerous foci of sebaceous differentiation (Hematoxylin-Eosin, Original Magnification 4×). (**B**) Immunohistochemical preparation with antibody anti-PRAME that shows the sebaceous cells positive with an immunoscore of intensity of 3+/4+ and a high percentage cells positivity of 2+/3+ (Immunohistochemical for PRAME, Original Magnification 4×). (**C**) Details of the previous histological picture showing the sebaceous cells positive for PRAME in a neoplastic aggregate of sebaceous carcinoma (Immunohistochemistry for PRAME, Original Magnification 10×). (**D**) Example of immunohistochemical preparation nearly totally negative, or only focal positive, for PRAME (Immunohistochemistry for PRAME, Original Magnification 10×).

**Table 1 jcm-11-06936-t001:** Clinical, topographical and surgical features of the SC included and studied in this paper. Note the number in the brackets represent the value of composite score of immunostaining.

Number of Case	Age	Gender	Topography	Treatment	PRAME Immunoexpression	WHO Grade
1	89	M	Upper Eyelid	Surgery + re-excision	Negative (3)	I
2	76	F	Lower Eyelid	Surgery + re-excision	Positive (6)	I
3	78	M	Eyelid	Surgery + re-excision	Negative (2)	II
4	92	M	Scalp	Surgery	Positive (5)	I
5	83	M	Scalp	Surgery	Negative (2)	II
6	96	F	Left preauricular	Surgery	Negative (3)	II
7	68	F	Right auricular	Surgery	Negative (3)	II
8	63	M	Not reported	Surgery + re-excision	Positive (4)	I
9	70	M	Forehead	Surgery + re-excision	Negative (3)	III
10	67	M	Left subscapularis	Surgery + re-excision	Positive (6)	I
11	70	M	Chest	Surgery + re-excision	Negative (2)	II
12	98	F	Right leg	Surgery	Positive (4)	I
13	76	F	Forehead	Surgery	Negative (2)	II
14	29	F	Chest	Surgery	Negative (2)	II
15	70	F	Lower eyelid	Surgery	Positive (5)	I
16	61	M	Forehead	Surgery + re-excision	Negative (3)	III
17	79	M	Neck	Surgery	Negative (2)	II
18	67	F	Forehead	Surgery	Positive (4)	I
19	62	M	Upper eyelid	Surgery + re-excision	Negative (2)	II
20	88	M	Right cheekbone	Surgery	Negative (2)	II
21	86	F	Conjunctival	Surgery + re-excision	Negative (2)	III

## Data Availability

Not applicable.

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
