# Peer review of "PRAME Immuno-Expression in Cutaneous Sebaceous Carcinoma: A Single Institutional Experience"

_jcm, 2022, doi:10.3390/jcm11236936_

Round 1

Reviewer 1 Report

Table must include degree and strength of staining not only if it positive or negative. 

If I am not wrong, all the cases showed are within the skin. You claim intraocular sebaceous carcinoma. I think you mean ocular-type sebaceous carcinoma as they are in the eyelid.

You claim a different pattern of staining in grade I and II sebaceous carcinomas compared to grade III ones. Please, include in the text the deffinition of grade I, II and III. Please, include in the table the WHO degree, please, add the proper statistics in order to compare the pattern in grade I and II and the one in III with the p significance level. Please include in the results and in the text, how the staining pattern differ.

Author Response

Reviewer n’1: Table must include degree and strength of staining not only if it positive or negative.

Answer n’1: Dear Reviewer n’1, first of all thank you very much for your kind suggestions useful to improve the quality of our manuscript. We added, in the table, the value of “composite score” for immunostaining for PRAME and we, also, added, in Material and methods section, the criteria of assessment of this one.

Reviewer n’1: If I am not wrong, all the cases showed are within the skin. You claim intraocular sebaceous carcinoma. I think you mean ocular-type sebaceous carcinoma as they are in the eyelid.

Answer n’2: Thank you very much. It’s right. So, we corrected the diction “intraocular” with ocular-type/eyelid. Thank you so much.

Reviewer n’1: You claim a different pattern of staining in grade I and II sebaceous carcinomas compared to grade III ones. Please, include in the text the deffinition of grade I, II and III. Please, include in the table the WHO degree, please, add the proper statistics in order to compare the pattern in grade I and II and the one in III with the p significance level. Please include in the results and in the text, how the staining pattern differ.

Answer n’3: Thanks again dear Reviewer n’1. So, we added the WHO grade information in Table 1, the definition of grade I-III in order to WHO 2018 in the right section and, furthermore, we conducted the statistical analysis with a Mann-Whitney test for non-parametric values. Finally, we included our results in the text and in the discussion section. Thanks again for all.

Reviewer 2 Report

There is no abstract part of the article. In the abstract part, there is the text containing the rules of the journal. The abstract part seems to have been forgotten. Has this been overlooked?

What is HF abbreviation?

The authors should point out the small number of cases as a limitation. If there are criteria for inclusion and exclusion in the study, they should be explained in detail.

Has ethical approval and consent been obtained?

Re-evaluation can be made after the deficiencies are corrected.

Author Response

Reviewer n’2: There is no abstract part of the article. In the abstract part, there is the text containing the rules of the journal. The abstract part seems to have been forgotten. Has this been overlooked?

Answer n’1: Dear Reviewer n’2, first of all thank you very much for your kind suggestions useful to improve the quality of our manuscript. Sorry for this mistake, but there was a problem during the uploading process of our manuscript. We added the abstract. Sorry again.

Reviewer n’2: What is HF abbreviation?

Answer n’2: Thank you very much. It’s wrong. We corrected.

Reviewer n’2: The authors should point out the small number of cases as a limitation. If there are criteria for inclusion and exclusion in the study, they should be explained in detail.

Answer n’3: Thanks again dear Reviewer n’2. We added as limitation the small number of our cases and we added some details about the methods in the section. Thanks again for all.

Reviewer n’3: Has ethical approval and consent been obtained?

Answer n’4: yes, we confirm that informed consent and ethical approval were obtained before the submission of the manuscript. Thanks a lot.

Round 2

Reviewer 1 Report

Authors have addressed suggestions